# Clinical Value of Circulating Microribonucleic Acids miR-1 and miR-21 in Evaluating the Diagnosis of Acute Heart Failure in Asymptomatic Type 2 Diabetic Patients

**DOI:** 10.3390/biom9050193

**Published:** 2019-05-17

**Authors:** Mutaa Abdalmutaleb Al-Hayali, Volkan Sozer, Sinem Durmus, Fusun Erdenen, Esma Altunoglu, Remise Gelisgen, Pınar Atukeren, Palmet Gun Atak, Hafize Uzun

**Affiliations:** 1Department of Biochemistry, Yildiz Technical University, Istanbul 34220, Turkey; alhayalimuttaa@yahoo.com; 2Department of Physiology, Biochemistry and Pharmacology College of Veterinary Medicine, Mosul University, Mosul 09334, Iraq; 3Department of Biochemistry, Cerrahpasa Faculty of Medicine, Istanbul University-Cerrahpasa, Istanbul 34098, Turkey; durmus.sinem@gmail.com (S.D.); remisagelisgen@hotmail.com (R.G.); p_atukeren@yahoo.com (P.A.); huzun59@hotmail.com (H.U.); 4Istanbul Training and research Hospital, Department of Internal Medicine, Istanbul 34098, Turkey; fusunozerdenen@hotmail.com (F.E.); esmaaltunoglu@yahoo.com (E.A.); 5Department of Biochemistry, Faculty of Medicine, Istanbul Bilim University, Istanbul 34394, Turkey; palmetgun@gmail.com

**Keywords:** asymptomatic type 2 diabetes mellitus, acute heart failure, silent coronary artery disease, NT-proBNP, galectin-3, miRNA-1, miRNA-21

## Abstract

To investigate whether the circulating miR-1 (microRNA-1) and miR-21 expression might be used in the diagnosis of heart failure (HF) and silent coronary artery disease (SCAD) in asymptomatic type 2 diabetes mellitus (T2DM) patients and to explore the relationship of these miRs with N-terminal pro-brain natriuretic peptide (NT-proBNP) and galectin-3. One hundred thirty-five consecutive patients with T2DM and 45 matched control subjects were enrolled in the study. This study consisted of the following four groups: control group (mean age: 60.23 ± 6.27 years, female/male (F/M): 23/22); diabetic group (DM) (mean age: 61.50 ± 5.08, F/M: 23/22); DM + SCAD group (mean age: 61.61 ± 6.02, F/M: 20/25); and DM + acute HF group (mean age: 62.07 ± 5.26 years, F/M: 20/25). miR-1 was downregulated in the DM, CAD + DM and HF + DM groups by 0.54, 0.54, and 0.12 fold as compared with controls, respectively. The miR-1 levels were significantly lower in HF + DM than DM with 0.22 fold changes (*p* < 0.001); and in patients with CAD + DM group with 0.22 fold changes (*p* < 0.001). Similarly, miR-21 was overexpressed in patients with DM, CAD + DM, and HF + DM with 1.30, 1.79 and 2.21 fold changes as compared with controls, respectively. An interesting finding is that the miR-21 expression was significantly higher in the HF + DM group as compared with the CAD + DM group; miR-1 was negatively correlated with NT-proBNP (*r* = −0.891, *p* < 0.001) and galectin-3 (*r* = −0.886, *p* < 0.001) in the HF + DM group; and miR-21 showed a strongly positive correlation with (*r* = 0.734, *p* < 0.001) and galectin-3 (*r* = 0.764. *p* < 0.001) in the HF + DM group. These results suggest that the circulating decreased miR-1 and increased miR-21 expression are associated with NT-proBNP and galectin-3 levels in acute HF + DM. Especially the miR-21 expression might be useful in predicting the onset of acute HF in asymptomatic T2DM patients. The miR-21 expression is more valuable than the miR-1 expression in predicting cardiovascular events of acute HF and the combined analysis of miR-21 expression, galectin-3, and NT-proBNP can increase the predictive value of miR-21 expression.

## 1. Introduction

Type 2 diabetes mellitus (T2DM) is a chronic disease and its prevalence has been rapidly rising in the middle- and low-income countries. Type 2 DM is known to have a negative influence on the presentation, severity, prognosis, and also prevalence of coronary artery disease (CAD) [1]. Type 2 DM and heart failure (HF) also commonly accompany each other in clinical practice. Years ago, HF was noted to be a complication of diabetes [2]. Increasing numbers of older patients with diabetes, and their improved survival from cardiovascular events will undoubtedly see a massive increase in patients with both diabetes and HF. Accurate diagnosis of HF is very important because of the high morbidity and mortality. A quarter of those with chronic HF have diabetes and over 40% of these patients are hospitalized with worsening HF [1,3].

The natriuretic peptides (NP) are vasoactive peptide hormones which are mainly secreted from heart tissue as an endocrine gland. There are four different groups of NPs identified to date, namely A–D type. N-terminal pro-brain natriuretic peptide (amino acids 1–76, NT-proBNP) is a prohormone. When it is cleaved from the molecule, natriuretic peptide (BNP) is released. BNP and NT-proBNP are reported to be useful diagnostic biomarkers for the prognosis in patients who have asymptomatic left ventricule (LV) dysfunction and different degrees of congestive heart failure (CHF) [4]. Measuring the levels of circulating BNP and NT-proBNP has been recommended both for the diagnosis and the management of HF [5,6,7,8]. However, NT-proBNP represents a more useful biomarker for the diagnosis and risk stratification of patients who have chronic HF because it is more stable in serum after blood collection and has a longer half-life. Even in the absence of HF, elevated circulating NT-proBNP levels have also emerged as a serologic marker for the assessment of cardiovascular disease [5,6,7,8]. Moreover, NT-proBNP has been shown to be a marker for subclinical atherosclerosis in T2DM patients [9] and elevated circulating NT-proBNP is suggested to be a strong predictor of cardiovascular mortality [10].

Galectin-3 is one of the 14 members of the lectin family and it is a pleiotropic protein which is encoded by a single gene (*LGALS3*). It binds various β-galactosides via its carbohydrate recognition domain (CRD), and reveals several biological effects. The CRD which consists of approximately 130 amino acids takes place in the pathophysiology of HF [11]. Galectin-3 also plays an important role in inflammation, tissue repair, including fibrogenesis. It also has a role in cardiac ventricular remodeling which is an important hallmark in HF [12,13,14]. In a recent meta-analysis, galectin-3 was validated as a biomarker with independent prognostic value for mortality and HF rehospitalization [15]. It is not clear if systemic diseases such as DM affect the predictive value of these HF biomarkers. It was found that DM has the least influence on the predictive power of certain HF biomarkers, such as galectin-3 and NT-proBNP [16].

MicroRNAs (miRNAs) are small non-coding RNAs, which is approximately 22 nucleotides in length. Many studies have implied that miRNAs are released into extracellular fluids. MicroRNAs exhibit hormone-like activities because extracellular miRNAs may be delivered by their target cells and they may act as autocrine, paracrine, and/or endocrine regulators to modulate cellular activities [17]. Extracellular miRNAs play a very crucial role in many biological processes such as cancer development, immune system, epithelial-to-mesenchymal transition and fibrosis, in various types of cardiac diseases, and in DM [18].

An asymptomatic patient with T2DM has been shown 20–35% prevalence of silent CAD (SCAD), which could lead to myocardial ischemia, adverse cardiac events, and a poor prognosis outcome [19]. Acute HF is a life-threatening emergency and there is a high percentage of DM among acute HF patients. DM patients with acute HF have a poor prognosis. DM has a negative effect on the prognosis of HF. Screening for DM and prevention of progressive cardiac injury is particularly clinically important. All hospitalized patients with acute HF should be screened for diabetes [20]. Expressions of miR-1 and miR-21 are dysregulation in HF [21]. Additionally, there is an inverse relationship between the miR-1 expression and the NT-proBNP levels in patients in the New York Heart Association (NYHA) class II/III. The upregulation of miR-21 correlates with galectin-3 levels. These miRNAs might be used in addition to the major conventional tests being used to evaluate HF in diabetic patients.

Patients with T2DM had a poor prognosis due to SCAD and acute HF, and therefore it is important to define any early predictors of SCAD and acute HF in patients with T2DM. In addition, there is not adequate information in the literature regarding the relationship between SCAD and acute HF patients with T2DM and miR-1 and miR-21 expressions. Therefore, this study aimed to investigate whether the circulating miR-1 and miR-21 expressions might be used in the diagnosis of SCAD and in acute HF with asymptomatic T2DM patients and without diabetes and to explore the relationship of these miRNAs with NT-proBNP and galectin-3.

## 2. Materials and Methods

### 2.1. Subjects

The protocol was approved by the local Ethics Comittee of the Istanbul Education and Research Hospital (No: 1031, date: 7 July 2017) and was conducted in accordance with Declaration of Helsinki. All subjects were of Turkish descent. All subjects gave their informed consent for inclusion before they participated in the study. Pregnant women, active infection, acute renal failure, hepatic, rheumatic, malignant or endocrine diseases, subarachnoid haemorrhage, chronic lung diseases, acute and chronic pulmonary embolism, smokers, individuals with a history of chronic alcohol consumption, and subjects who were taking certain drugs such as hepatotoxic drugs (antituberculous, antiepileptic) or oral contraceptive pills were excluded from the study. All subjects were classified into four different groups:Control group: A total of 45 healthy subjects who did not have any endocrine, vascular, cardiac or inflammatory diseases were chosen for the control group (mean age: 60.23 ± 6.27 years, female/male (F/M): 23/22). An oral questionnaire was conducted with the subjects and none of our subjects declared that they had a family history of diabetes. They did not have diabetes or glucose intolerance as confirmed by an oral glucose tolerance test (OGTT).DM group: Patients with T2DM (mean age: 61.50 ± 5.08, F/M: 23/22) who were diagnosed according to the American Diabetes Association (ADA) guidelines [22] were included in this study. All of the diabetic patients were being treated for diabetes with insulin (20%) and/or metformin (80%).DM + CAD group: A total of 45 diabetic patients (mean age: 61.61 ± 6.02, F/M: 20/25) with coronary artery disease were enrolled in our study. All of the diabetic patients were being treated for diabetes with insulin (25%) and/or metformin (75%). A total of 86% of the diabetic patients in this group had hypertension and they were being treated with beta blockers (48%), thiazide (28%) and/or angiotensin-converting enzyme (ACE) inhibitors (14%). The diabetic patients with dyslipidemia (75%) were taking antihyperlipidemic drugs such as statins. Patients with 2 or 3 vascular occlusion were selected as the CAD + DM group.Acute HF + DM group: A total of 45 patients (mean age: 62.07 ± 5.26 years, F/M: 20/25) with acute HF were studied.

The diagnostic criteria of acute HF were recommended according to the American Heart Association (AHA) Guidelines. The exclusion criteria were: patients with T2DM known HF, chronic obstructive pulmonary disease (COPD), pulmonary artery embolism and/or deep venous thrombosis, or who used to take anticoagulant drugs in the past three months or received a blood transfusion recently, patients with a history of malignancy, cognitive dysfunction, mental illness, systemic disease.

### 2.2. Coronary Angiography

Patients underwent coronary angiography using the Seldinger technique. Three-dimensional digital subtraction angiography (3D-DSA) examinations were performed with femoral catheterization with a DSA system (Philips Allura Xper FD20, Veenpluis, Netherlands). The coronary angiograms were read by expert cardiologists, as well as the clinical diagnosis, and the laboratory results. The cardiologists recorded the location and extent of luminal narrowing for 15 segments of the major coronary arteries [23]. Patients were classified as having CAD if a stenosis of 50% or greater was found in at least one of the segments. Patients without CAD were defined as having less than 50% stenosis in all of the segments. A composite cardiovascular score (0–75) was calculated based on determination of presence of stenosis on a scale of 0–5 of the 15 predetermined coronary artery segments.

### 2.3. Echocardiography

Each patient underwent a complete transthoracic echocardiographic (TTE) study using an ultrasound system located in adjacent echocardiography rooms. The device used was the General Electric Vivid S5 (GE Health Medical, Horten, Norway). The left ventricular ejection fraction (LVEF) was calculated according to Simpson’s method [24]. Heart failure was defined according to the clinical criteria of the Framingham Heart Study 6 and by 25% < LVEF < 35%, according to the bidimensional transthoracic Doppler echocardiography.

### 2.4. Sample Collection and Measurements

Fasting venous blood samples were drawn between 8 and 10 a.m. after the subjects fasted overnight (10–12 h). Blood samples were drawn from the brachial veins in brachial fossa and placed into plain tubes (K2-EDTA anticoagulated whole blood) and anticoagulant free tubes. The samples were centrifuged for 10 min at 4000 rpm at 4 °C. Biochemical tests were performed immediately. For the determination of other parameters, serum aliquots were frozen and stored at –80 °C immediately until they were required for further analysis.

Serum miRNA was extracted from serum samples using an EXTRACTME miRNA KIT (BLIRT, Gdańsk, Poland). All isolation protocols were conducted according to the manufacturers’ instructions, without further modifications. The concentrations and purities of RNA were estimated using a NanoDrop spectrophotometer (ThermoFisher Scientific, Wilmington, DE 19810, USA), and values of ~2.0 were considered indicative of relatively pure RNA. To estimate the expression miRNA levels two-step reverse transcriptasa (RT)-PCR assay was used. Normalization was calculated according to the determined miRNA levels.

Firstly, the miRNA samples were transcribed into completentary DNA (cDNA) using the high-capacity cDNA RT Kit (ThermoFisher Scientific) and RT oligos specific to the miRNAs identified in Table 1 (SUARGE, Istanbul, Turkey), then the cDNA synthesis was performed. Then cDNAs were amplified and the expressions of miR-1 and mir-21 analyzed using the StepOnePlus real-time PCR system (Applied Biosystems, Carlsbad, CA, USA) using specific AMPLIFYME SYBR Universal Mix (BLIRT, Gdańsk, Poland). RNU44 (TaqMan Small RNA Controls from Applied Biosystems) was used as a small RNA endogenous control. The relative expression levels of miRNA-1 and mir-21 were calculated using the 2^−ΔΔCT^ method. Each sample was tested in triplicate.

### 2.5. Measurement of Plasma NT-proBNP Concentrations

The NT-proBNP levels were analyzed (K2-EDTA anticoagulated whole blood) using time resolved florescence assay on the AQT90 FLEX immunoassay analyzer (Radiometer, Copenhagen, Denmark). The results were expressed as pg/mL. The intra- and inter-assay coefficients of variation (CVs) were determined to be 4.1% and 5.2%, respectively.

### 2.6. Measurement of Plasma Galectin-3 Concentration

Serum galectin-3 levels were also assayed using a sandwich ELISA kit (Human galectin-3 kit, Cat. No. E-EL-H1470, Elabscience Biotechnology Co., Wuhan, Hubei, China). The galectin-3 results were expressed as ng/mL. The lowest level of galectin-3 that could be detected by this assay was 0.10 ng/mL. The intra- and inter-CV were determined to be 6.6% and 7.5%, respectively.

Biochemical rutin parameters (glucose, total cholesterol, high density lipoprotein (HDL)-cholesterol, low density lipoprotein (LDL) cholesterol, triglyceride, creatinine, uric acid) were determined using the spectrophotomrtric methods (Roche Cobas İntegra 400, Roche Diagnostics Ltd., Mannheim, Germany). HbA1c determination was based on HPLC (Variant Turbo II, Bio-Rad Laboratories, Inc., Hercules, CA, USA).

### 2.7. Statistical Analysis

For statistical analyses, SPSS v. 22.0 (IBM, Armonk, NY, USA) was used. Continuous variables were tested for normal distribution using the Shapiro–Wilk test. Results for normally distributed continuous variables were expressed as means ± standard deviations, and we used the unpaired Student’s *t*-test to compare mean values. Between-group comparisons of distributions were performed using the Mann–Whitney U test and Wilcoxon’s signed-rank sum test. Correlations among continuous variables were assessed using Spearman’s rank correlation coefficient (*r*). Categorical variables were expressed as numbers (percentages) and were compared using Fisher’s exact test. To evaluate the expression levels of miRNAs circulating within the two groups, we decided to use Student’s *t*-test for two independent groups. The receiver operating characteristic (ROC) analysis was used to determine the separation power of the parameters. As a result of the ROC analysis, cut-off points were determined by using the Youden Index. To determine the risk of having the values above the cut-off value, the risk analysis was performed and the OR (odds ratio) values were obtained. The positive predictive values of the combinations according to the cut-off points for the miR-21, NT-proBNP, and galectin-3 parameters were calculated and *p*-values < 0.05 were considered statistically significant.

## 3. Results

### 3.1. Patient’s Characteristics

One hundred and thirty five diabetic patients was divided into three subgroups: patients with diabetes mellitus only (DM) (*n* = 45, mean age 61.50 ± 5.08), patients with CAD together with diabetes mellitus (CAD + DM) (*n* = 45, mean age 61.61 ± 6.02), and patients with heart failure with diabetes (HF + DM) (*n* = 45, mean age 62.07 ± 5.26). There were no statistically significant differences in terms of age and sex between these groups and also between the control and these groups (Table 2). The diastolic blood pressure (DBP) values of the control individuals were significantly lower than the DM, the CAD + DM, and the HF + DM groups (for each *p* < 0.001). There was no difference between the other groups in terms of DBP. The systolic blood pressure (SBP) values of control subjects were significantly lower than the DM, the CAD + DM, and the HF + DM groups (for each *p* < 0.001). In the DM group, SBP was significantly lower than the CAD + DM group (*p* < 0.05). In the CAD + DM group, SBP was significantly lower than the HF + DM group (*p* < 0.001), (Table 2).

There was no correlation found between the EF and miRNAs and the cardiovascular risk factors, lipid parameters, systolic and diastolic pressure, body mass index (BMI), creatinine, and uric acid. Similarly, there was no association between the cardiovascular risk factors (lipid parameters, systolic and diastolic pressure, BMI) and miRNAs in the CAD + DM patients. It was not shown in the table, but there was no correlation between the number of vascular occlusion and cardiovascular risk factors (lipid parameters, systolic and diastolic pressure, BMI) and miRNAs in the CAD + DM group.

### 3.2. Biochemical Findings and Expression of MicroRNAs

The Serum NT-proBNP and galectin-3 levels were found to be increased significantly in the DM, the CAD + DM and the HF + DM groups with respect to the control group, respectively (for NT-proBNP: control vs. DM *p* < 0.001, control vs. CAD + DM *p* < 0.01, control vs. HF + DM *p* < 0.001, DM vs. CAD + DM *p* < 0.01, DM vs. HF + DM *p* < 0.001, CAD + DM vs. HF + DM *p* < 0.05; for galectin-3 *p* < 0.001 for all comparisons) (Table 2).

The serum miR-1 levels were significantly lower in patients from the DM, the CAD + DM and the HF + DM groups than the control groups, with 0.54-, 0.54- and 0.12-fold changes, respectively (for each group *p* < 0.001). The miR-1 levels were significantly lower in patients from the HF + DM group than in patients with DM with 0.22-fold changes (*p* < 0.001); and in patients from the CAD + DM group with 0.22-fold changes (*p* < 0.001) (Figure 1).

On the contrary, the serum miR-21 levels were significantly upregulated in patients from the CAD + DM and HF + DM groups than the control groups, with 1.79-, and 2.21-fold changes, respectively (for each group *p* < 0.001). The miR-21 levels also were significantly higher in patients from the HF + DM group than in patients with DM with 1.70-fold changes (*p* < 0.001)., and in patients from the CAD + DM group with 1.24-fold changes (*p* < 0.01) and in patients from the CAD + DM group than in patients with DM with 1.37-fold changes (*p* < 0.001) (Figure 2).

Figure 3 shows that miR-1 correlations with NT-proBNP and galectin-3 in the DM, CAD + DM and HF + DM groups. In the DM group, miR-1 was found to be positively correlated with NT-proBNP (*r* = 0.419, *p* < 0.01). In other groups, miR-1 was found to be negatively correlated with NT-proBNP (for CAD + DM *r* = −0.882, *p* < 0.001; for HF + DM *r* = −0.891, *p* < 0.001) and galectin-3 (for DM *r* = −0.371, *p* < 0.05; for CAD + DM *r* = −0.754, *p* < 0.001; for HF + DM *r* = −0.866, *p* < 0.001).

Figure 4 shows that miR-21 correlations with NT-proBNP and galectin-3 in the DM, CAD + DM and HF + DM groups. In contrast to miR-1, in all groups miR-21 was positively correlated with NT-proBNP and (for DM *r* = 0.893, *p* < 0.001; for CAD + DM *r* = 0.898, *p* < 0.001; for HF + DM *r* = 0.734, *p* < 0.001) and galectin-3 (for DM *r* = 0.782, *p* < 0.001; for CAD + DM *r* = 0.773, *p* < 0.001; for HF + DM *r* = 0.764, *p* < 0.001).

In the second part of the study, the ROC curve of serum biomarkers and miRNA fold-change levels were determined for different group combinations and cut-off points were determined (Figure 5). According to the results of the ROC analysis, miR-21, NT-proBNP, and galectin-3 might be good biomarkers to distinguish HF + DM (miR-21—sensitivity: 84.4%, specificity: 71.1%, *p* < 0.001; NT-proBNP—sensitivity: 100%, specificity: 100%, *p* < 0.001; galectin-3—sensitivity: 80.0%, specificity: 100%, *p* < 0.001) for miR-21, NT-proBNP, and galectin-3, respectively, from DM (Table 3). Although it is not as powerful as other parameters, the miR-1 levels were found to be statistically significant in differentiating HF + DM patients from DM according to the ROC analysis (sensitivity: 46.7%, specificity: 22.2%, *p* < 0.001, Table 3). Analysis results for other parameters are shown in Figure 5 and Table 3.

According to the cut-off values, the risk of the disease was calculated for those who had higher levels than the cut-off values (Table 3). When compared to DM, the serum miR-1 and miR-21 expression levels were found to be higher than 0.125- and 1.463-fold change, respectively, and increased the risk of developing HF + DM by 1.667 fold and 12.020-fold, respectively (Table 3). The results of the analysis for other parameters are shown in Table 3.

Compared to the control group; serum galectin-3 levels higher than 4.95 ng/mL were found to increase the risk of developing DM by 7.5 fold. In addition, when the DM and the CAD + DM groups are compared; serum galectin-3 levels higher than 7.15 ng/mL were shown to increase the risk of CAD + DM by 18.5 fold. And it is interesting that as compared with the CAD + DM and HF + DM groups; galectin-3 levels greater than 9.25 ng/mL were found to increase the risk of HF + DM by 7.6 folds. The serum NT-proBNP levels higher than 95.5 pg/mL increased the risk of developing DM by 6.18 folds (compared to the control); and higher than 4747 pg/mL increased the risk of HF + DM (according to CAD + DM) (Table 3).

In cases where the value of miR-21 was higher than 1.695 and the NT-proBNP value was higher than 4747 pg/mL, the probability of HF + DM (positive predictive value) was found as 95.2%. In cases where the value of miR-21 was lower than 1.695 and the NT-proBNP value was less than 4747 pg/mL, the probability of being only DM (positive predictive value) was found to be 66.7% (Table 4).

In cases where the value of miR-21 was higher than 1.695 and the galectin-3 value was higher than 9.25 ng/mL, the probability of HF + DM (positive predictive value) was found as 74.4%. In cases where the value of miR-21 was less than 1.695 and the galectin-3 value was less than 9.25 ng/mL, the probability of being only DM (positive predictive value) was found to be 51.5% (Table 5).

In the cases where the NT-proBNP value was higher than 4747 pg/mL and the galectin-3 value was greater than 9.25 ng/mL, the probability of HF + DM (positive predictive value) was found to be 100%. In the cases where the NT-proBNP value was lower than 4747 pg/mL and the galectin-3 value was less than 9.25 ng/mL, the probability of being only DM (positive predictive value) was found as 63.4% (Table 6).

In the cases where the value of miR-21 was higher than 1.695, the NT-proBNP value was 4747 pg/mL and the galectin-3 value was greater than 9.25 ng/mL, the probability of HF + DM (positive predictive value) was found to be 100%. In cases where the value of miR-21 was 1.695, the NT-proBNP value was lower than 4747 pg/mL, and the galectin-3 value was less than 9.25 ng/mL, the probability of being only DM (positive predictive value) was found to be 69.4% (Table 7).

## 4. Discussion

In this study, we report that miR-1 and miR-21 expressions were associated with the presence of DM. The miR-1 expression was found to be downregulated, while the miR-21 expression was overexpressed in all patients as compared with the control subjects. We noted several important differences among the DM, CAD + DM and HF + DM groups with respect to miR-1 and miR-21. First, among subjects with DM, miR-1 was the lowest in HF + DM. Second, the miR-21 expression is higher in the HF + DM group as compared with the CAD + DM group. Furthermore, miR-1 negatively correlated with the NT-proBNP and galectin-3 levels in the CAD + DM group and miR-21 showed stronger positive correlation with NT-proBNP and galectin-3. According to the results of the ROC analysis, miR-21 might be a good biomarker to distinguish HF + DM from DM. Our results underscore the importance of decreased miR-1 and increased miR-21 expression as a cardiovascular risk factor and suggest that miR-21 can be used as an early predictor of HF in asymptomatic T2DM patients.

DM is known to be a potent and prevalent risk factor for heart disease. Diabetic cardiomyopathy is an early complication of DM and is revealed with diastolic dysfunction followed by abnormalities in systolic function [25]. Due to the NYHA functional class, the mortality risk associated with HF patients is not explained fully with comorbidities (DM, anemia, and renal insufficiency) and treatment strategies [26]. It is not clear if systemic diseases such as DM have an effect on the predictive value of biomarkers for HF. In the current study, serum NT-proBNP and galectin-3 levels were the highest in HF + DM. In the study by Ballo et al. [26] they investigated the association between NT-proBNP and risk of cardiac events, in a population of asymptomatic diabetic patients who were enrolled in a primary care setting, and found that NT-proBNP levels added an independent and an incremental prognostic value for the prediction of the clinical outcome. The serum NT-proBNP levels were found to be increased in the DM, CAD + DM and HF + DM groups. Furthermore, the NT-proBNP levels were higher in the HF + DM group as compared to the CAD + DM group. Both the current and the other mentioned study [27,28,29,30] suggest that an elevated plasma NT-proBNP level in asymptomatic patients with DM should alert physicians for an increased risk of the cardiovascular events.

It is interesting that as compared with the CAD + DM and HF + DM groups, galectin-3 levels in CAD + DM was found lower than HF + DM. Tan et al. [31] investigated the relationship between serum galectin-3 and incident cardiovascular events and all-cause mortality in T2DM patients. According to the results of this study, serum galectin-3 was found to be related to the adverse outcomes in subjects with or without prevalent CAD, independent from the traditional cardiovascular risk factors. The results of galectin-3 in T2DM seems confusing because some studies claim that galectin-3 deficiency is associated with insulin resistance, and galectin-3 elicits a protective effect in T2DM by acting as a receptor for advanced glycation end products (AGEs) [32,33]. However, Li et al. [34] showed that in galectin-3 gene knock out mice which were fed with a high-fat diet, the development of insulin resistance was found to be significantly reduced. Furthermore, the results of this study also provided preliminary evidence that extracellular galectin-3 binds to the insulin receptor directly and attenuates downstream pathways, suggesting galectin-3 to be a novel targetable link between the insulin resistance and T2DM. Holmager et al. [35] suggested that glucose metabolism is associated with circulating galactin-3 in HF, as elevated levels were found in patients with DM and a relation with increasing HbA1c levels was also demonstrated. Both ours and the mentioned previous study [31,32,34,35] suggest that there is an urgent need to develop galectin-3 inhibitors that have a high oral bioavailability and a low toxicity profile to combat increased galectin-3 levels which are related with the developing HF in DM.

The circulating miR-1 which have anticardiac hypertrophic effects was found downregulated in all patients groups when compared with the control subjects in the present study. In addition, miR-1 expression was found to be decreased gradually in the DM, CAD + DM and HF + DM groups Furthermore miR-1 was negatively correlated with the NT-proBNP and galectin-3 levels in the CAD + DM and HF + DM groups and it especially showed the strongest correlation in the HF + DM group. Although not as powerful as the other parameters, miR-1 levels were still found to be statistically significant in differentiating HF + DM patients from DM according to the ROC analysis. Similar to our results, Sygitowicz et al. [21] demonstrated that miR-1 was significantly downregulated. Downregulation of the expression of miR-1 was correlated with the increase of serum NT-proBNP concentration in patients with symptomatic HF in the NYHA class II/III. As no patients with acute myocardial infarction were included in this study, the miR-1 expression pattern was not influenced by the presence of acute ischaemia or myocardial necrosis, which could potentially increase the expression of this type of miR. In the present study, as compared with DM, the serum miR-1 expressions were found to be higher than 0.125-fold change and increased the risk of developing HF + DM by 1.667-fold. According to the results of our study, the miR-1 expression together with the NT-proBNP and galectin-3 levels may be useful in predicting the onset of HF in asymptomatic T2DM patients. There are controversial findings concerning the importance of miR-1 in heart disease. Plasma miR-1 was upregulated in patients with acute myocardial infarction (AMI)-HF [36,37]. Ai et al. [38] showed that miR-1 levels were significantly higher in the plasma of AMI patients compared with non-AMI subjects and the levels were dropped to normal values on discharge following the medication. Increased circulating miR-1 was not associated with age, gender, blood pressure, DM or the established biomarkers for AMI. Restoration of miR-1 gene expression was a potential novel therapeutic strategy to reverse pressure-induced cardiac hypertrophy and prevent maladaptive cardiac remodeling. Tomaniak et al. [39] found that in symptomatic HF patients with LVH, galectin-3 concentrations and miR-1 expressions were correlated with anatomic changes of the left ventricle. Karakikes et al. [40] suggested that the restoration of miR-1 gene expression was a potential novel therapeutic strategy to reverse pressure-induced cardiac hypertrophy and prevent maladaptive cardiac remodeling. The miR-1 expression was suggested to have therapeutic potential either via being used to target specific genes, or via becoming a therapeutic target itself [41].

The roles of miR-21 in cardiac diseases are controversial [42,43,44,45,46,47,48,49]. The reason for this is probably because miR-21 plays different roles in different cell types. As well, miR-21 is only present in interstitial cells and correlates with collagen expression in the heart [46]. In the present study, serum miR-21 expression was upregulated in patients in the CAD + DM and HF + DM groups and found higher than the control groups, with 1.79-, and 2.21-fold changes, and miR-21 levels also were significantly higher in patients from the HF + DM group than in patients with DM with 1.70-fold changes. No correlation was found between the EF and the miRNAs. However, miR-21 correlated with NT-proBNP and galectin-3 in the DM, CAD + DM and HF + DM groups. Similar to our results, Sygitowicz et al. [21] reported that overexpression of miR-21 was seen in all patients, independent of HF severity and overexpression of miR-21 was correlated significantly with galectin-3 levels. Furthermore, according to the results of the ROC analysis, miR-21 might be a good biomarker to distinguish HF + DM from DM. Contrary to our results, Tomaniak et al. [39] found significant downregulation of miR-21 associated with the increase of LVEDD in symptomatic HF and that miR-21 was upregulated in fibroblasts with high glucose treatment and exerted its harmful effects [43]. Dai et al. [50] demonstrated that miR-21 exerted its protective role directly in cardiac myocytes and encouraged further development of cardiac specific overexpression of miR-21 therapy toward cellular tropism. Circulating some miRs have also been suggested as promising diagnostic biomarkers in patients with T2DM. However, the results of miR-21 in T2DM are controversial [51,52,53,54]. Other studies showed that serum miR-21 level was not associated with T2D, however, its expression was significantly downregulated in serum of obese diabetic and non-diabetic subjects [51,52]. Zampetaki et al. [53] reported that plasma miR-21 levels were downregulated in T2DM patients when compared to non-diabetic subjects. Similar to our results, an upregulation of circulation miR-21 was reported in subjects with T2D as compared with prediabetic subjects [54]. The reason of the discrepancy among different studies remains unclear. A possible explanation regarding the divergence among the studies might be due to the differences in the source of samples (plasma vs. serum) or the differences in study populations [51]. When the results of predictive analysis are taken into consideration, we also believe that miR-21, NT-proBNP, and galectin-3 as activity panel indicator will be useful especially for HF + DM.

### Limitations of the Study

However, this study has some limitations which should be noted. Basically, at the beginning of the study our aim was to investigate both miR-3p and miR-5p, as miR-1-5p and miR-1-3p are biologically different in terms of sequence, stability, and functionality. However, this does not indicate that only one strand is functional, both strands could be functional, however, for economic reasons we were only able to examine the miR-1-5p. Thus, it would be useful to examine miR-1-3p, as well as to compare the functionality of miR-1-5p and miR-1-3p, so we are planning to analyze miR-1-3p in our next study.

## 5. Conclusions

We found that miR-1 and miR-21 are the main profibrogenic miRs, and they are independently associated with the presence and severity (decreased EF) of acute HF in T2DM. The miR-1 expression together with NT-proBNP and galectin-3 may be useful in predicting the onset of HF in asymptomatic T2DM patients. In addition, miR-21 may be a good biomarker to distinguish HF + DM from DM. These findings have led to the use of miR-1 and miR-21 as the circulating potential biomarkers for therapeutic prospects of HF. The roles of miR-1 and miR-21 in the pathogenesis of symptomatic HF in T2DM need to be determined in a large-scale prospective study.

## Figures and Tables

**Figure 1 biomolecules-09-00193-f001:**
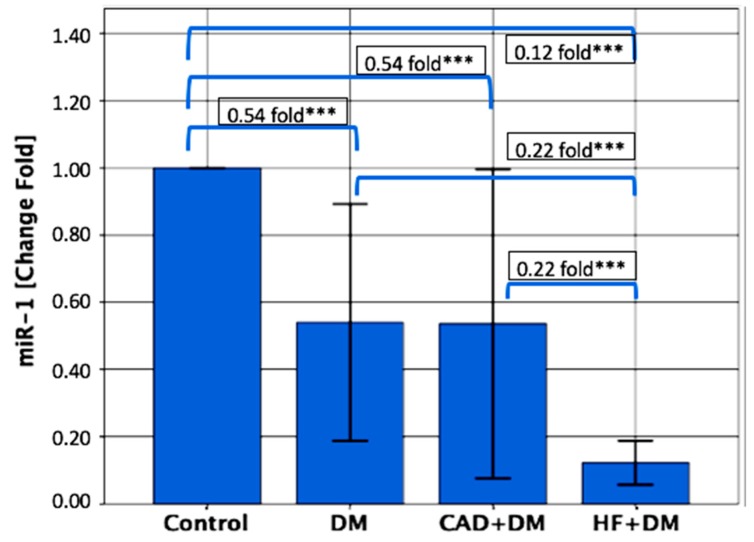
Relative expression levels of miR-1. The miR-1 levels were significantly lower in patients from the DM, CAD + DM and HF + DM groups than in the control groups, with 0.54-, 0.54- and 0.12-fold changes, respectively. The miR-1 levels were significantly lower in patients from the HF + DM group than in patients with DM with 0.22-fold changes; and in patients from the CAD + DM group with 0.22-fold changes. Data are presented as fold-change derived from mean 2^−ΔΔCT^ method. *** *p* < 0.001.

**Figure 2 biomolecules-09-00193-f002:**
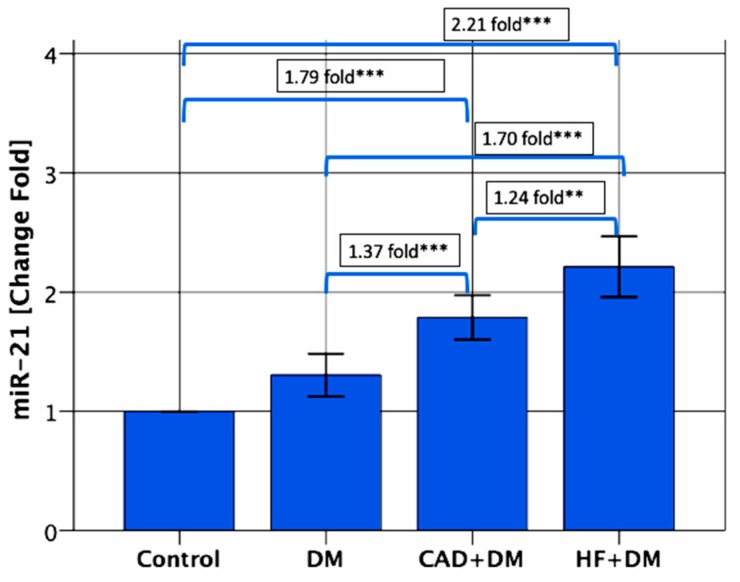
Relative expression levels of miR-21. The miR-21 levels were significantly higher in patients from the CAD + DM and HF + DM groups than the control groups, with 1.79-, and 2.21-fold changes, respectively. The miR-21 levels were significantly higher in patients from the HF + DM group than in patients with DM with 1.70-fold changes, and in patients from the CAD + DM group with 1.24-fold changes, and in patients from the CAD + DM group than in patients with DM with 1.37-fold changes. Data are presented as fold-change derived from mean 2^−ΔΔCT^ method. ** *p* < 0.01 *** *p* < 0.001.

**Figure 3 biomolecules-09-00193-f003:**
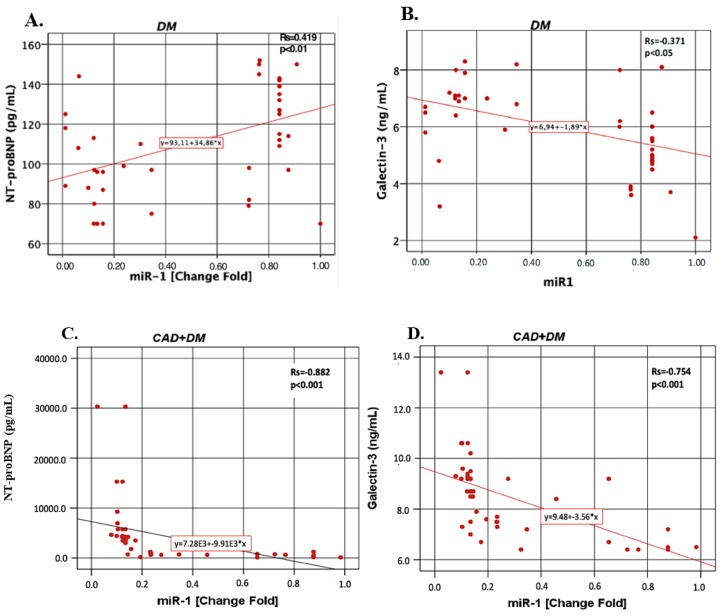
The miR-1 correlations with NT-proBNP and galectin-3 in the DM, CAD + DM and HF + DM groups. Data are presented as fold-change derived from mean 2^−ΔΔCT^ method. Rs: Spearman’s rank correlation coefficients (*r*) (**A**) miR-1 vs. NT-proBNP in DM group; (**B**) miR-1 vs. galectin-3 in DM group; (**C**) miR-1 vs. NT-proBNP in CAD + DM group; (**D**) miR-1 vs. galectin-3 in CAD + DM group; (**E**) miR-1 vs NT-proBNP in HF + DM group; (**F**) miR-1 vs galectin-3 in HF + DM group.

**Figure 4 biomolecules-09-00193-f004:**
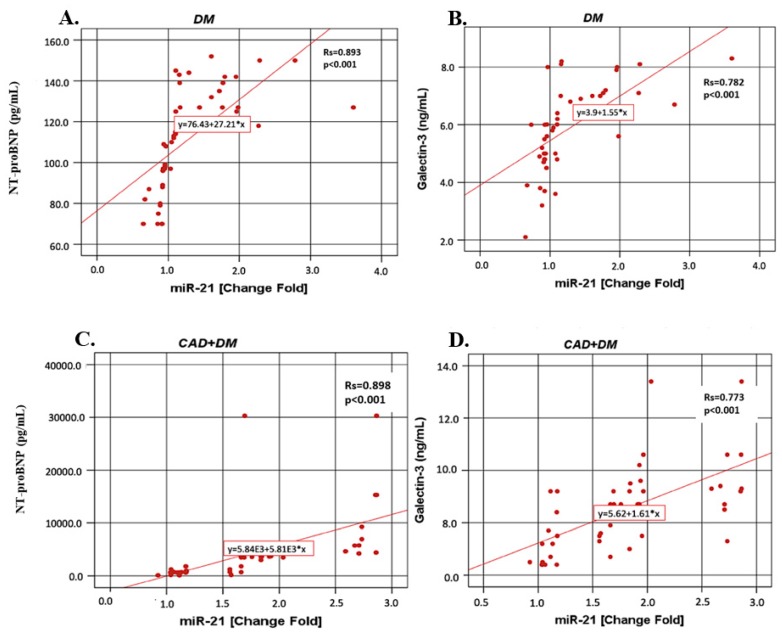
The miR-21 correlations with NT-proBNP and galectin-3 in the DM, CAD + DM and HF + DM groups. Data are presented as fold-change derived from mean 2^−ΔΔCT^ method. Rs: Spearman’s rank correlation coefficients (*r*). (**A**) miR-21 vs. NT-proBNP in DM group; (**B**) miR-21 vs. galectin-3 in DM group; (**C**) miR-21 vs. NT-proBNP in CAD + DM group; (**D**) miR-21 vs galectin-3 in CAD + DM group; (**E**) miR-21 vs. NT-proBNP in HF + DM group; (**F**) miR-21 vs galectin-3 in HF + DM group.

**Figure 5 biomolecules-09-00193-f005:**
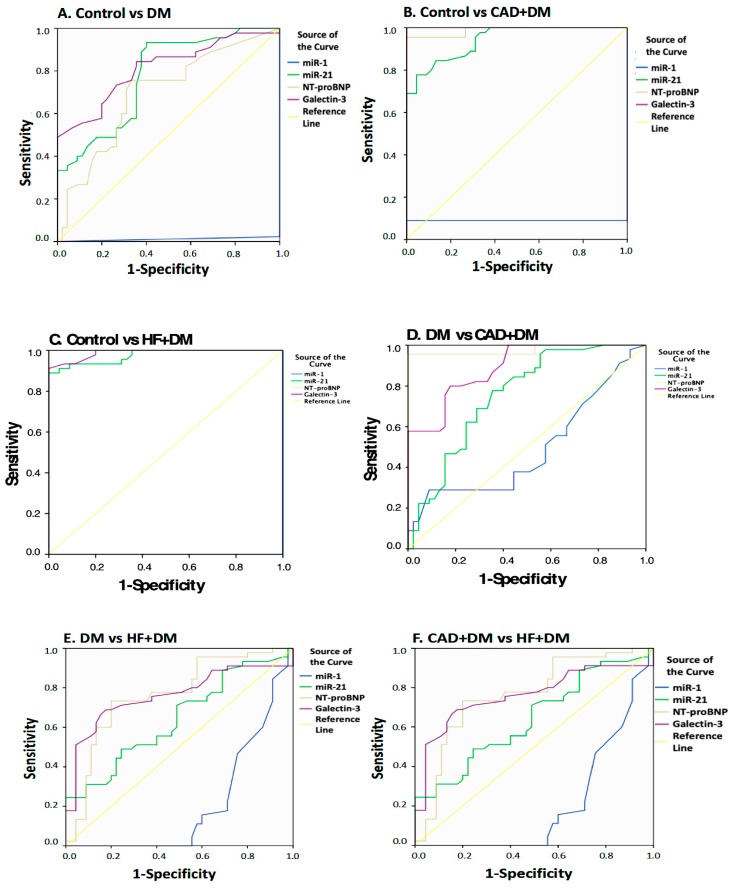
The ROC analysis of laboratory findings for different groups: (**A**) Control vs. DM groups, (**B**) Control vs. CAD + DM groups, (**C**) Control vs. HF + DM groups, (**D**) DM vs. CAD + DM groups, (**E**) DM vs. HF + DM groups, and (**F**) CAD + DM vs. HF + DM vs. CAD + DM groups.

**Table 1 biomolecules-09-00193-t001:** Reverse transcription oligos specific to miRNAs.

Primers	Sequence
hsa-miR-1 RT	5′-GAAAGAAGGCGAGGAGCAGATCGAGGAAGAAGACGGAAGAATGTGCGTCTCGCCTTCTTTCATGGGCAT-3′
hsa-miR-21 RT	5′-GAAAGAAGGCGAGGAGCAGATCGAGGAAGAAGACGGAAGAATGTGCGTCTCGCCTTCTTTCTCAACATC-3′
RNU44 RT	5′-GAAAGAAGGCGAGGAGCAGATCGAGGAAGAAGACGGAAGAATGTGCGTCTCGCCTTCTTTCAGTCAGTT-3′
hsa-miR-1 Forward	5′-GCAACATACTTCTTTATATGCCCAT-3′
hsa-miR-21 Forward	5′-GCGGTAGCTTATCAGACTGATGT-3′
RNU44 Forward	5‘-CCTGGATGATGATAAGCAAATG-3′
Universal reverse	5′-CGAGGAAGAAGACGGAAGAAT-3′

RT: reverse transcription, miRNA: microRNA.

**Table 2 biomolecules-09-00193-t002:** General characteristics and biochemical parameters of subjects.

	Control Group (*n* = 45)	Diabetes Mellitus (*n* = 45)	DM + CAD (*n* = 45)	DM + HF (*n* = 45)
Sex (F/M)	23 (51%)/22 (49%)	23 (51%)/22 (49%)	20 (44%)/25 (56%)	20 (44%) /25 (56%)
Age (years)	60.23 ± 6.27	61.50 ± 5.08	61.61 ± 6.02	62.07 ± 5.26
BMI (kg/m^2^)	23.33 ± 1.39 ^a***,b***,c***^	29.86 ± 3.18	29.62 ± 2.63	29.21 ± 1.66
DM duration (years)	-	6.83 ± 2.50 ^b*,c***^	7.96 ± 2.73 ^a*,c*^	10.50 ± 4.17
DBP (mmHg)	75.32 ± 5.07 ^a***, b***, c***^	83.88 ±6.15	84.93 ± 7.41	84.09 ± 7.80
SBP (mmHg)	113.50 ± 6.22 ^a***, b***, c***^	136.88 ± 5.85 ^b*^	130.71 ± 14.95 ^c***^	141.36 ± 16.40
Glucose (mg/dL)	83.03 ± 9.85 ^a***,b***,c***^	210.03 ± 77.69 ^b*,c***^	164.89 ± 54.90 ^c***^	92.64 ± 4.35
HbA1c	4.74 ± 0.47 ^a**,b***,c***^	10.23 ± 9.67 ^c*^	7.46 ± 1.42 ^c***^	5.63 ± 0.73
Total cholesterol (mg/dL)	169.73 ± 13.90 ^a***,c***^	231.73 ± 60.36 ^b***,c*^	186.39 ± 32.88	198.36 ± 27.81
HDL cholesterol (mg/dL)	51.03 ± 8.87 ^b***,c***^	52.63 ± 14.58 ^b***,c***^	41.39 ± 6.75	40.07 ± 8.49
LDL cholesterol (mg/dL)	93.58 ± 11.43 ^a***,b***,c***^	143.65 ± 41.33 ^b**,*c^	116.00 ± 26.57	119.55 ± 34.19
Triglyceride	99.53 ± 16.24 ^a***,b***,c***^	196.98 ± 135.86	153.39 ± 61.47	149.43 ± 46.30
Creatinine (mg/dL)	0.82 ± 0.16 ^c***^	0.86 ± 0.33 ^c**^	1.17 ± 0.76	1.19 ± 0.49
Uric Acid	6.08 ± 0.88	5.69 ± 1.66	6.30 ± 2.31	6.50 ± 1.33
hsCRP (mg/L)	0.47 ± 0.26 ^c*^	1.79 ± 5.64	1.79 ± 3.19	3.33 ± 5.92
NT-proBNP (pg/mL)	95.10 ± 24.00 ^a***,b**,c***^	111.93 ± 25.23 ^b**,c***^	4556.44 ± 6533.47 ^c*^	8764.71 ± 6863.97
Galectin-3 (ng/mL)	4.27 ± 1.13 ^a***,b***,c***^	5.92 ± 1.48 ^b***,c***^	8.50 ± 1.63 ^c***^	10.68 ± 2.81

DM: diabetes mellitus; DBP: diastolic blood pressure; SBP: systolic blood pressure; HbA1c: Glycated hemoglobin; hsCRP: high sensitivity C-reactive protein; NT-proBNP: N-terminal pro-brain natriuretic peptide, CAD: coronary artery disease; HF: heart failure; F/M: female/male. BMI: body mass index; HDL: high density lipoprotein; LDL: low density lipoprotein. ^a^ vs. DM, ^b^ vs. DM + CAD, ^c^ vs. DM + HF. * *p* < 0.05, ** *p* < 0.01, *** *p* < 0.001.

**Table 3 biomolecules-09-00193-t003:** The ROC analysis of laboratory findings for different groups and risk assesment according to cut-off values.

**Control vs. DM**						
**Variables**	**AUC**	***p*-Value**	**Sensitivity**	**Specificity**	**Cut-off Value**	**OR for Cut-off Values (95%CI**)
miR-1	0.011	0.000	-	-	-	-
miR-21	0.777	0.000	0.933	0.600	0.840	21.000 (5.641–78.173) ***
NT-proBNP	0.694	0.002	0.756	0.667	95.5	6.182 (2.464–15.512) ***
Galectin-3	0.812	0.000	0.733	0.733	4.95	7.563 (2.971–19.251)***
**Control vs. CAD + DM**						
**Variables**	**AUC**	***p*-Value**	**Sensitivity**	**Specificity**	**Cut-off value**	**OR for cut-off Values (95%CI**)
miR-1	0.089	0.000	-	-	-	-
miR-21	0.944	0.000	0.800	0.911	1.130	38.875 (10.338–124.488) ***
NT-proBNP	0.988	0.000	0.956	1.000	167.5	X
Galectin-3	1.000	0.000	1.000	1.000	6.200	X
**Control vs. HF + DM**						
**Variables**	**AUC**	***p*-Value**	**Sensitivity**	**Specificity**	**Cut-off Value**	**OR for Cut-off Values (95% CI**)
miR-1	0.000	-	-	-	-	-
miR-21	0.974	0.000	0.889	1.000	1.223	X
NT-proBNP	1.000	0.000	1.000	1.000	210.500	X
Galectin-3	0.988	0.000	0.911	1.000	6.700	X
**DM vs. CAD + DM Groups**						
**Variables**	**AUC**	***p*-Value**	**Sensitivity**	**Specificity**	**Cut-off value**	**OR for Cut-off Values (95% CI**)
miR-1	0.492	0.063	-	-	-	Not significant
miR-21	0.755	0.000	0.778	0.667	1.168	7.0 (2.742–17.867) ***
NT-proBNP	0.976	0.000	0.956	1.000	165.5	X
Galectin-3	0.891	0.000	0.800	0.822	7.150	18.5 (6.428–53.245) ***
**DM vs. HF + DM Groups**						
**Variables**	**AUC**	***p*-Value**	**Sensitivity**	**Specificity**	**Cut-off Value**	**OR for Cut-off Values (95%CI**)
miR-1	0.194	0.000	0.467	0.222	0.125	1.667 (1.175–2.363) ***
miR-21	0.834	0.000	0.844	0.711	1.463	12.020 (4.320–33.460) ***
NT-proBNP	1.000	0.000	1.000	1.000	208.5	X
Galectin-3	0.933	0.000	0.800	1.000	8.40	X
**CAD + DM vs. HF + DM Groups**						
**Variables**	**AUC**	***p*-Value**	**Sensitivity**	**Specificity**	**Cut-off Value**	**OR for Cut-off Values (95%CI**)
miR-1	0.207	0.000	0.467	0.244	0.125	1.619 (1.137–2.306) ***
miR-21	0.640	0.022	0.711	0.511	1.695	2.570 (1.078–6.144) *
NT-proBNP	0.764	0.000	0.733	0.800	4747.00	11.0 (4.108–29.454) ***
Galectin-3	0.761	0.000	0.711	0.756	9.250	7.608 (2.981–19.417) ***

* *p* < 0.05; *** *p* < 0.001; AUC: area uder curve; OR: odds Ratio; X: OR could not be calculated because there were no values above or below the cut-off value in the groups; CI: confidnce interval.

**Table 4 biomolecules-09-00193-t004:** Positive predictive values of combinations based on cut-off for miR-21 and NT-proBNP.

Cut-off for miR-21 and NT-proBNP	DM		CAD + DM		HF + DM	
	*n*	%	*n*	%	*n*	%
miR-21 < 1.696 and NT-proBNP < 4748	34	66.7	15	29.4	2	3.9
miR-21 > 1.695 and NT-proBNP < 4748	11	26.2	21	50.0	10	23.8
miR-21 < 1.696 and NT-proBNP > 4747			8	42.1	11	57.9
miR-21 > 1.696 and NT-proBNP > 4747			1	4.8	20	95.2

**Table 5 biomolecules-09-00193-t005:** Positive predictive values of combinations based on cut-off for miR-21 and galectin-3.

Cut-off for miR-21 and Galectin-3	DM		CAD + DM		HF + DM	
	*n*	%	*n*	%	*n*	%
miR-21 < 1.696 and galectin-3 < 9.26	34	51.5	22	33.3	10	15.2
miR-21 > 1.695 and galectin-3 < 9.26	11	42.3	12	48.0	2	8.0
miR-21 < 1.696 and galectin-3 > 9.25			1	25.0	3	75.0
miR-21 > 1.695 and galectin-3 > 9.25			10	26.3	28	73.7

**Table 6 biomolecules-09-00193-t006:** Positive predictive values of combinations based on cut-off for NT-proBNP and galectin-3.

Cut-off for NT-proBNP and Galectin-3	DM		CAD + DM		HF + DM	
	*n*	%	*n*	%	*n*	%
NT-proBNP < 4748 and galectin-3 < 9.26	45	63.4	25	35.2	1	1.4
NT-proBNP > 4747 and galectin-3 < 9.26			9	42.9	12	57.1
NT-proBNP < 4748 and galectin-3 > 9.25			11	50.0	11	50.0
NT-proBNP > 4747 and galectin-3 > 9.25					21	100.0

**Table 7 biomolecules-09-00193-t007:** Positive predictive values of combinations based on cut-off for miR-21, NT-proBNP, and galectin-3.

Cut-off for miR-21, NT-proBNP, and Galectin-3	DM		CAD + DM		HF + DM	
	*n*	%	*n*	%	*n*	%
miR-21 < 1.696 and NT-proBNP < 4748 and galectin-3 < 9.26	34	69.4	14	28.6	1	2.0
miR-21 > 1.695 and NT-proBNP < 4748 and galectin-3 < 9.26	11	50.0	11	50.0		
miR-21 < 1.696 and NT-proBNP > 4747 and galectin-3 < 9.26			8	47.1	9	52.9
miR-21 > 1.696 and NT-proBNP > 4747 and galectin-3 < 9.26			1	33.3	2	66.7
miR-21 < 1.696 and NT-proBNP < 4748 and galectin-3 > 9.25			1	50.0	1	50.0
miR-21 > 1.695 and NT-proBNP < 4748 and galectin-3 > 9.25			10	50.0	10	50.0
miR-21 < 1.696 and NT-proBNP > 4747 and galectin-3 > 9.25					2	100.0
miR-21 > 1.695 and NT-proBNP > 4747 and galectin-3 > 9.25					18	100.0

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
