# Peer review of "Clinical Value of Circulating Microribonucleic Acids miR-1 and miR-21 in Evaluating the Diagnosis of Acute Heart Failure in Asymptomatic Type 2 Diabetic Patients"

_biomolecules, 2019, doi:10.3390/biom9050193_

Reviewer 1 Report

Overall, the responses are satisfactory. 

Author Response

First, we would like thank the reviewer for the helpful comments

Reviewer 2 Report

The authors' response on the removal of miR-1 related discussion from manuscript is not acceptable. Even though the authors are correct regarding the lack of specific description on 3p or 5p for miR-1 in literature, most of the commercially available primers are measuring the miR-1-3p.  The primer information is available from Taqman, Qiagen (Exiqon), and some other companies.  There is no miR-1-5p primer available from most of the vendors; therefore, it is unlikely that the miR-1 related information in literature is on miR-1-5p. The level of miR-1-5p is very low, the authors can check the alignment results from miRBase, there is very little sequencing read mapped to the -5p region compared to -3p (http://www.mirbase.org/cgi-bin/get_read.pl?acc=MI0000437). 

Author Response

First of all, thank you very much for your constructive comments and contributions. Within the scope of our study, indeed we wanted to examine both miR-1-3p and miR-1-5p. Before completing our work and also after your comments, we requested additional budget by contacting our university to also examine miR-1-3p. However, the reply which we were able to get as return was “Your budget for this study is limited, we recommend that you to work with this parameter the next time when you work.”. Therefore, unfortunately, we could not analyze miR-1-3p because we could not find a budget.  Also, we would like to express that its available to find miR-1-5p primers commercially (For Ex: ABMGOOD, ThermoFisher Sci,) and we have also attached two recent literatures related to the detecton of miR-1-5p, and in our study we prefered to detect miR-1-5p. However, your invaluable contributions have encouraged us to analyze miR-1-3p in our next study. Thus, we will have the opportunity to show the difference between the two strands (miR-1-3p and miR-1-5p) in the literature by referring to this study.

Round  2

Reviewer 2 Report

None

This manuscript is a resubmission of an earlier submission. The following is a list of the peer review reports and author responses from that submission.

Round  1

Reviewer 1 Report

The responses are satisfactory.

Reviewer 2 Report

1.       In the revised manuscript, it is the acute heart failure that the authors are focusing. However, there is no information about acute heart failure in the introduction. It is the congestive heart failure was mentioned in the introduction. Both acute and chronic heart failure can be ruled out by BNP or NT-proBNP, however, the cut-off thresholds used for both conditions are not the same. Authors need to give an introduction on acute heart failure in relationship with T2DM - this is the focus on of their study.

2.       Authors added the introduction of SCAD, however, most of the description is on HF and DM. Based on objective of the study: “To investigate whether the circulating (microRNA-1) miR-1 and miR-21 expression might be used as an early predictor for heart failure (HF) and silent coronary artery disease (CAD) in asymptomatic type 2 diabetes mellitus (T2DM) patients …”, authors should focus on acute heart failure and T2DM; SCAD and T2DM… Write the context in a broad range and leads into the need/aim of the study.

3.       What are the inclusion/exclusion clinical criteria for acute HF+DM group?

4.       For point 8 comments to reviewer: Authors explained that “parallel work was carried out using the EXTRACTME miRNA KIT (BLIRT) kit and miRNeasy Serum / Plasma Kit (QIAGEN) in their previous study”. Please cite the reference if this has been published. Also, please present the parallel data for both kits – justify how an extraction kit meant for tissue and cell lines can be used for serum samples. This is to ensure the data is reproducible.

5.       For point 9 comments to reviewer: Authors explained that “Total RNA isolation was performed as the first step in our study. A ratio of 1.7~2.0 is generally accepted as “pure” for total RNA.” This does not reply to my question. It is understood that a ratio of 1.7-2.0 is accepted for total RNA, but this is only applicable to RNA extracted from tissue or cell line samples, but not for RNA extracted from serum/plasma samples. Please provide the nanodrop readings of the extracted RNA for the serum samples used in this study, are the samples nanodrop readings for A260/280 and A260/230 fall in the range of 1.7-2.0?

6.       Figure 5- duplicated figure 5A and 5B but missing figure 5C and 5D- this should not have happened in a revised manuscript! Authors should check and proofread thoroughly before submission.

7.       For point 20 comments to reviewer: Define OR cut-off value? Is Median or mean used?

8.       For point 21 comments to reviewer: Authors mentioned Youden index is used. Please include this in statistical analysis. Define the cut-off value.

9.       Line 355: proBNP or NT-proBNP?

10.   Please explain how the cut-off points for miR-21, NT-proBNP and Galectin-3 are determined in the new positive predictive tables 4-7. Based on Table 6, using combined NT-proBNP and galection-3 achieved 100% of HF+DM, adding miR-21 is not needed. The 100% predictive value for HF+DM in Table 7 is due to the combination of NT-proBNP and Galectin-3. Mir-21 does not provide additional value as a predictor.

11.   What is the reason to exclude two subjects from HF+DM group for Table 4, 5 and 7?

12.   The title states that miR-1 and miR-21 as an early predictor of HF in T2DM. However, the evidence provided is too weak to support this claim. miR-1 failed to be a predictor and miR-21 is neither a strong predictor.

Reviewer 3 Report

The authors addressed some of the concerns.  However, the authors should remove the literature review relates to miR-1 in the Discussion section, which was requested in prior comments.  The authors were assessing the level of miR-1-5p which is different from miR-1-3p in the literature.  Because of the sequence difference the miR-1-3p and miR-1-5p may have different functional impact.  The authors should also state that they are assessing the level of miR-1-3p and must remove the miR-1 related information in the manuscript.